# Latent Microsporidia Infection Prevalence as a Risk Factor in Colon Cancer Patients

**DOI:** 10.3390/cancers14215342

**Published:** 2022-10-29

**Authors:** Fernando Redondo, Carolina Hurtado-Marcos, Fernando Izquierdo, Carmen Cuéllar, Soledad Fenoy, Yanira Sáez, Ángela Magnet, Lorena Galindo-Regal, Natalia Uribe, Manuel López-Bañeres, Ana Isabel Jiménez, Antonio Llombart-Cussac, Carmen Del Águila, Juan Carlos Andreu-Ballester

**Affiliations:** 1Facultad de Farmacia, Universidad San Pablo-CEU, CEU Universities, Urbanización Montepríncipe, 28660 Boadilla del Monte, Spain; 2Department of Microbiology and Parasitology, Complutense University, 28040 Madrid, Spain; 3Molecular Biologist, Laboratory of Molecular Biology and Research Department, Arnau de Vilanova University Hospital, FISABIO Foundation, 46015 Valencia, Spain; 4Department of General and Digestive Surgery, Arnau de Vilanova University Hospital, 46015 Valencia, Spain; 5Pathology Department, Arnau de Vilanova University Hospital, 46015 Valencia, Spain; 6Medical Oncology Department, Arnau de Vilanova Hospital, Catholic University of Valencia, 46015 Valencia, Spain; 7FISABIO Foundation and Research Department, Arnau de Vilanova University Hospital, c/San Clemente 12, 46015 Valencia, Spain

**Keywords:** microsporidia, *Encephalitozoon* sp., *Enterocytozoon bieneusi*, colon cancer, IFAT, ELISA

## Abstract

**Simple Summary:**

Microsporidia infection has been related to the malignant process of epithelial cells. We found a high prevalence of microsporidia in the intestinal tissues of patients with Colon Cancer (CC) vs tissues of healthy subjects. This observation could suggest a relationship between microsporidia and the etiopathogenesis of CC.

**Abstract:**

Microsporidia are opportunistic intracellular parasites, generating serious pathology in individuals with a compromised immune system. Infection by microsporidia inhibits p53 and Caspase 3, proteins involved in apoptosis and the cell cycle, which are vital in the malignant process of epithelial cells. The presence of microsporidia in the intestinal tissues of 87 colon cancer (CC) patients and 25 healthy controls was analyzed by real-time PCR and an immunofluorescence antibody test. Anti-*Encephalitozoon* antibodies were analyzed in serum samples by ELISA (enzyme linked immunosorbent assay). In 36 (41.3%) CC cases, microsporidia infections were identified in their tissues vs. no cases among control subjects (*p* < 0.0001). An increase in IgG and IgE anti-*Encephalitozoon* antibodies was found in patients with CC, which would demonstrate continuous and previous contact with the parasite. The high prevalence of microsporidia in tissues and the seroprevalence in patients with CC suggest a relationship between microsporidia and the etiopathogenesis of CC.

## 1. Introduction

Many risk factors have been linked to Colon Cancer (CC), including hereditary, environmental, and inflammatory syndromes that affect the gastrointestinal tract [1]. Loss of genomic stability in epithelial cells of the intestine is one of these mechanisms since it facilitates the appearance of mutations in genes that encode proteins functionally associated with the control of chromosomal stability, repair of defective DNA, cell cycle control or tumor suppression. It is outstanding that p53 gene alteration is one of the most frequent genetic mutations in human cancer [2]. 

It is also essential to know the factors that set in motion the molecular processes that lead to the malignant transformation of cells. In this sense, it has been identified that specific infectious agents can induce cancer in humans. *Helicobacter pylori*, hepatitis B and C, papilloma and, Epstein-Barr viruses have been linked to the development of stomach, liver, and nasopharyngeal cancer, respectively. Human immunodeficiency virus (HIV) and human herpesvirus 8 (HHV-8) are linked to Kaposi’s sarcoma and lymphomas [3,4]. Among parasites, *Trichomonas vaginalis* has been associated with cervical and prostate cancers; *Toxoplasma gondii* is related to ocular tumors, meningiomas, leukemias, and lymphomas; and *Plasmodium* spp. have been associated with Burkitt’s lymphomas [5,6,7,8]. The intracellular intestinal parasite *Cryptosporidium parvum* has been shown to induce digestive adenocarcinoma, even with low doses of inoculum, in an experimental murine model. It has also been described that cryptosporidiosis is associated with colon carcinomas in patients with human immunodeficiency virus [9,10]. *Schistosoma haematobium* is associated with urinary bladder cancer and *Opisthorchis viverrini* and *Clonorchis sinensis* with cholangiocarcinoma [11]. Infection with *Schistosoma japonicum* is particularly interesting in our study as it has been associated with the development of colorectal cancer. The tumorigenic mechanisms that have been studied involve chronic inflammation, mutations of p53 by the action of toxins, and inhibition of the activity of the immune system, promoting the accumulation of myeloid-derived suppressor cells in lymphoid tissues and the tumor microenvironment [12]. Recently, an increased risk of opportunistic *Blastocystis* sp. infection was associated with colorectal cancer [13]. 

Microsporidia are obligate intracellular parasites that cause severe infections in immunosuppressed subjects. Nevertheless, it has also been detected in healthy immunocompetent subjects [14]. Pathogenic effects range from diarrheal processes to ocular disease, sinusitis, myositis, encephalitis, and disseminated infection [15]. The presence of intestinal microsporidia has been investigated in patients with cancer undergoing chemotherapy, both in children and adults. Several authors have found a prevalence of 4% to 21% [16,17,18,19,20], and only one study found a very high prevalence of 69.9% in cancer patients undergoing chemotherapy treatment [21]. 

In previous research carried out by our group, it was observed that *Encephalitozoon* microsporidia modulated the immune response by regulating the apoptosis induction pathway and the cell cycle, inhibiting the activation of the apoptotic protein Caspase-3 and the transcription of the universal protective tumor suppressor protein, p53 [22]. On the other hand, it has been described that *Encephalitozoon intestinalis* infection increased host cellular mutation frequency in mice, indicating a possible connection between microsporidiosis and cancer induction [23].

Furthermore, previous studies have discussed the relationship between the etiopathogenesis of Crohn’s disease (CD) and microsporidia. In Crohn’s, a deficiency of CD3+CD8+γδ T cells has been described, located in the mucous epithelium. These cells are essential for the defense against infections and tumors [24,25]. These studies are critical and relevant since individuals with CD are at risk of developing CC [26]. Moreover, patients with newly diagnosed treatment-*naïve* infiltrating colonic adenocarcinoma have a significant decrease in γδ T cells compared to healthy control volunteers [27].

Based on the above considerations, the aim of this study was to investigate the presence of microsporidia in tissues of patients with CC without previous specific anti-tumor treatment.

## 2. Materials and Methods

### 2.1. Ethics Statement 

The Hospital Research Ethics Committee approved the study, and each patient signed an informed consent document. This epidemiological survey was carried out in compliance with fundamental ethical principles, including those reflected in the Charter of Fundamental Rights of the European Union and the European Convention on Human Rights and its supplementary protocols. All participants attested to their involvement in this clinical research study through written informed consent. The consent was evaluated and approved by the Research Ethics Committee of the Arnau de Vilanova Hospital (CEIm), following the recommendations of the Spanish Bioethics Committee, the Spanish legislation on Biomedical Research (Law 14/2007, of July 3), and Personal Data Protection (Spanish Law 3/2018 and European Law UE676/2018). These laws define that access to the clinical record for judicial, epidemiological, public health, research, or educational purposes carries an obligation to keep the patient’s identification data separated from clinical and healthcare data; therefore, anonymity is ensured.

### 2.2. Subjects of Study and Clinical Samples

Eighty-seven newly diagnosed colon cancer patients were recruited at the Arnau de Vilanova Hospital in Valencia (Spain). Colon biopsies were obtained from both tumor and normal peritumoral tissue through a surgical procedure from 87 patients, (Table 1). Tissues were also collected from 25 control subjects with no pathological findings in the colon cancer screening program. Tissue samples were separated into two equal parts. One was kept at −80 °C to extract the DNA, and the other part was paraffinized to perform the histological sections analyzed by IFAT. For this reason, before performing DNA extraction of the sample, tissue imprints were made on new slides. These imprints subsequently served to complete the immunofluorescence tests. 

For the serological study, seventy-two serum samples were obtained from CC patients and 25 from healthy controls.

Finally, the following exclusion criteria were taken into account, both in patients with CC and in controls: absence of infectious, inflammatory, or autoimmune disease, known immunodeficiency, and a lack of specific anti-tumor or immunosuppressive treatment in CC patients; and the application of some vaccine for at least one year.

### 2.3. Immunofluorescence Antibody Test (IFAT)

#### 2.3.1. Deparaffination and Imprints

Deparaffination of tissue sections was carried out in several dissolvents for ten minutes in each one of the following steps: xylol, xylol-ethanol, ethanol absolute, ethanol 90%, ethanol 70%, and distilled water [25]. Both deparaffined tissues and imprints were aired dry and fixed with methanol-acetone. The contours of the sample were marked with a DAKOPEN^®^ marker (Biolabs) to contain the IFAT solutions.

#### 2.3.2. Assay

For the assay, two primary monoclonal antibodies (MAb) were used. The MAb 2C2 (murine origin and isotype IgG2a) recognizes the wall’s exospore and the developing states of the *Encephalitozoon* species [28]. The MAb 6E52D9 (murine origin and isotype IgG2a) recognizes the exospore of the wall of *Enterocytozoon bieneusi* [29]. Monoclonal antibodies (undiluted supernatant) were added to each well (tissue or imprint) on each slide and incubated at 37°C for 60 min in a wet chamber. Slides were washed three times in distilled water and then were air-dried at room temperature. 

The second antibody Fluorescein Isothiocyanate conjugated (Sigma, St. Louis, MO, USA), was diluted following the manufacturer’s instructions. The slides were incubated again at 37 °C for 60 min, washed three times in distilled water, and air dried at room temperature in the dark. The mounting liquid (PVA/Dabco) was added to the slides, covered by a cover slip, and examined at 40× with a Nikon immunofluorescent microscope. Positive controls were included using species homologous to microsporidia spores of each monoclonal antibody used. The samples and the positive controls were stored in the dark at 4 °C until the reading.

### 2.4. Encephalitozoon Cuniculi Antigen and Determination of Specific Antibodies (ELISA)

*E. cuniculi* antigen was obtained from *E. cuniculi* (USP-A1) spores following the Aguila et al. protocol [30]. Briefly, spores were disrupted using glass beads, 2.5% SDS, and 2% mercaptoethanol. Soluble antigens were obtained from the supernatant after centrifugation. Protein content was determined by the Bradford method and adjusted to 0.8 mg/mL to coat ELISA plates. Duplicate dilutions of human serum at 1/100 in PBS-Tween, containing 0.1% BSA, were added and incubated. Horseradish peroxidase (HRP) conjugated goat anti-human total immunoglobulins (Ig’s), IgG, IgM, or IgA (Biosource International, Camarillo, CA) were used. For IgE determination, test serum was added in duplicate at a 1/2 dilution. A murine monoclonal antibody against a human IgE chain (IgG1k, E21A11, INGENASA, Madrid, Spain) was added and incubated, followed by a goat anti-mouse IgG1 (gamma) HRP conjugate (CALTAG Laboratories, Burlingame, CA, USA) [31,32].

### 2.5. DNA Extraction and Detection of Microsporidia by Multiplex Real-Time PCR 

DNA from the 174 colon biopsies (87 patients) was extracted using the Dneasy Blood and Tissue kit, Qiagen^®^. Real-Time PCR was carried out with the SYBR Green reagent following the protocol of Polley et al. [33], modified by Andreu-Ballester et al. [23]. The following primers were utilized: MsRTF (5′-CAGGTTGATTCTGCCTGACG-3′) and MsRTR (5′-CCATCTCTCAGGCTCCCTCT-3′), which ensure amplification of *Encephalitozoon intestinalis*, *Encephalitozoon cuniculi*, *Encephalitozoon hellem,* and *Enterocytozoon bieneusi* SSU-rRNA sequences. Forward and reverse primers (10 pmol each) were used in 20 µL reactions containing 5 µL template DNA. The cycling conditions were 95 °C for 10 s, 60 °C for 20 s, and 72 °C for 20 s. The assay was set up in duplicate for each sample and run for 35 cycles.

### 2.6. Data Analysis

A Mann-Whitney U test was used to study the quantitative variables. Odds Ratio’s (OR) were used to study the relation between positive and negative microsporidia with CC patients and healthy controls. The level of significance was taken as a P-value of less than 0.05 (bilateral contrast). Data were analyzed using the statistical software SPSS, version 19.

## 3. Results

### 3.1. Prevalence of Microsporidia in Tissues

Thirty-six (41.3%) patients of the 87 studied with CC (by PCR and IFAT) had microsporidia in their tissues (Table 2 and Figure 1A). Twenty-six (30%) of these patients with CC were positive by PCR vs. none in the tissues of control subjects (25 samples with negative PCR), Odds Ratio (OR) = 36.1 (IC 95% = 2.1–613.0) *p* < 0.013. Thus, the probability of finding microsporidia in colon tissue from patients with CC is over 36 times higher than in tissues from healthy subjects. 

### 3.2. Determination of Specific Antibodies

The levels of anti-*E. cuniculi* Ig’s, IgG, IgM, and IgA were determined in the serum of 72 patients with CC, and in 64 sera for IgE. A significant increase in IgE and IgG anti-*E. cuniculi* was observed in patients with CC vs healthy subjects (*p* < 0.01 and *p* < 0.05, respectively) (Figure 1B).

Serum that presented an optical density (OD) above the mean plus the standard deviation of the control group was considered as positive when comparing the CC and control groups. Firstly, 42.2% of the serum from patients with CC showed a positive result for IgE, followed by IgG (25%) and IgA (22.2%). Significant differences among percentages in CC and controls were observed in the case of IgG and IgE isotypes (*p* < 0.05). The lowest levels were found in the cases of IgM and Ig’s with percentages of 9.7% and 11.1% of positive serum, respectively (Figure 1C). 

Of the 72 serums from CC patients studied, 40 (55.6%) were positive for some of the specific isotypes analyzed by ELISA. Of the 40 CC sera (55.6%) that were positive for some class of immunoglobulin, 35 were analyzed in tissue samples. Of these 35 CC patients, 12 (34.5%) were positive by PCR or IFAT in some tissue. 

If we consider the patients who had a positive result on PCR and/or IFAT, 58.3% of them were positive for some of the isotypes studied by ELISA. On the one hand, from the serum of the patients who had a positive result in PCR, 55.6% of them showed anti-*E. cuniculi* antibodies. On the other hand, of the serum samples that had a positive result on IFAT, 60% of them showed anti-*E. cuniculi* antibodies. Finally, in the patients that were both positive via PCR and IFAT, 62.5% of them showed some positive antibody isotype by ELISA against *E. cuniculi.*

Furthermore, a significant increase in specific IgE in patients with CC and a positive PCR result in tissues versus patients with CC and a negative PCR results in tissues (*p* = 0.007) was found.

## 4. Discussion

This report is the first study investigating the relationship of microsporidia with CC and examining the presence of this parasite in surrounding healthy and tumor tissues. A prevalence of 41.3% of microsporidia in the colon tissue of patients with CC was found, compared to 0% of microsporidia in the colon tissues of healthy subjects. The prevalence rate of microsporidia infection in immunocompetent subjects ranges from 0% to 50% in different countries, although most studies are based on the detection of spores in feces with a few in urine [19,34,35,36,37].

To date, only one study has analyzed a control group of small bowel biopsies from healthy subjects, finding a 0% prevalence of microsporidia [25], similar to the findings of our current study. Additionally, finding microsporidia in feces or urine does not imply that the parasite is present in the tissues, causing infection and disease.

There are about 1400 described species of microsporidia, of which 14 have been shown to infect humans. In the present study, primers that detect *Encephalitozoon* sp. (*E. hellem, E. intestinalis,* and *E. cuniculi*) and *Enterocytozoon bieneusi* were used. For this reason, there could be some species present in the tissues that we were unable to detect, so the prevalence of microsporidia in patients with CC could be higher. 

Research looking into the molecular basis of the possible origin of cancer due to microsporidia infection showed that the parasite was capable of inhibiting apoptosis through different mechanisms. In this “in vitro” study, the mechanisms by which microsporidia regulate and modulate apoptosis were studied. It was observed that the cells infected with *Encephalitozoon* sp. did not form the exclusion product caspase-3 (which is an apoptosis pathway). This fact would indicate that microsporidia can modulate and regulate this activation pathway, suggesting that the parasite could modulate, or even inhibit, the apoptotic response of the host cell [22].

On the other hand, it has been described in patients with intestinal infections that the levels of hydrogen peroxide, advanced protein oxidation products, and malondialdehyde are higher than in uninfected people [38]. This result would indicate more significant oxidative stress in the organism and, therefore, a greater probability of damage to the cell’s genetic material, thus increasing the risk of cancer [39]. Another study revealed that microsporidia infection increased the frequency of nuclear mutations and that the increase in these mutations depended on the viable spores. The conclusion of this study showed that microsporidia infection is genotoxic to the host cell. This means that, either indirectly or directly, it would induce damage to cellular DNA, producing mutations that could result in cancer development in the patient in the long term [23]. 

Concerning the present work and specifically the detection of microsporidia spores in the analyzed tissue samples, the combination of immunological and molecular diagnostic techniques was shown to be a helpful tool that allows for the obtainment of highly reliable results. Due to the complexity of the diagnosis of this group of parasites, the complementarity of different techniques is necessary to reach a coherent and accurate result. Therefore, in the present work it was decided to use immunological diagnostic methods using monoclonal antibodies (IFAT) and molecular approaches to reinforce the results obtained. Both techniques have advantages and disadvantages, but the combination of both has allowed for consistent interpretation and justification of the obtained results.

The results obtained in the tissue samples studied by IFAT and real-time PCR reflected four different sets of results. The first group includes the samples that were positive in both IFAT and PCR. The positivity of both techniques would show a correct correlation, confirming the result obtained. The second group of samples with double negativity in IFAT and PCR would reflect the absence of the parasite in the samples studied. The third group with a negative IFAT and a positive PCR would be due to the lack of sensitivity of the IFAT and/or low parasite load of the sample that the PCR has detected due to its greater sensitivity. Finally, the fourth group with a positive IFAT and a negative PCR would be a result related to the possible presence of inhibitors in the sample, degradation of the genomic material of the sample, or the extrusion of the spores with the loss of parasite DNA. It should be noted that the two monoclonal antibodies used in the present work have allowed for the detection of the four most prevalent species in humans, as well as the possibility of identifying foci of infection or intracellular states of the parasite indicative of an active infection in the patient.

Regarding the study of seroprevalence and characterization of specific antibodies, the increase in anti-*Encephalitozoon* IgE in patients with CC compared to healthy subjects is noteworthy. Also, the significant increase in patients with CC and positive PCR vs. patients with negative PCR should be considered. The relationship of microsporidia with Crohn’s disease through PCR has already been proven, as well as the increase in anti-*Encephalitozoon* IgE for the same patients [25]. 

Moreover, Crohn’s disease could be a precursor to CC. Both pathologies have another common factor: the deficiency of γδ T cells and high microsporidia prevalence [25,27]. Therefore, there are a number of factors that can justify its relationship with the appearance of colon cancer. The deficiency of γδ T cells would favor infection by microsporidia, which in turn, by different mechanisms [22], would induce malignant transformation, which exacerbates the deficient defense against the tumor by the γδ T cells [27]. According to this hypothesis, the sequence of events includes first a γδ T cell deficit, then infection by microsporidia, and lastly the development of cancer. It seems essential to study the relationship of microsporidia infection with lymphocyte populations in CC patients. This study could give a clue as to who is the inducer of the tumor process.

Despite this, the high levels of specific IgG found in CC patients would indicate continuous and previous contact with the parasite. This could be due to successive exposures or a chronic infection due to an immune deficit. These results are consistent when compared with the results obtained for IgM levels. In this case, very low levels were obtained that would indicate low rates of primary infection in these patients. These values would confirm that the presence of the parasite predates the cancer diagnosis and would result in the microsporidia-cancer association. In the case of IgA anti-*Encephalitozoon* antibodies, slightly high levels were obtained that would confirm the continuous exposure to the parasite antigens. This could stimulate the production of this isotype, associated with the defense of the intestinal mucosa and characteristic of the response against enteric parasites.

Finally, it is necessary to highlight the high levels of IgE observed that reach a positivity rate close to 50%. This result shows that the parasite is active within the tumor lesion, continuously stimulating the immunoglobulin class change with its antigens. In this sense, it is necessary to mention that the production of this immunoglobulin is associated with the extrusion mechanism of the spores [40]. The presence of specific IgE was independent of the microsporidia species, appearing in patients in whom PCR had detected the presence of the parasite in both tumor and healthy tissue. This shows that anti-*Encephalitozoon* IgE is a highly sensitive and specific marker for microsporidia in tissues.

## 5. Conclusions

Microsporidium infection was found in the tissues of more than 40% of CC patients compared to 0% of healthy subjects. Significantly higher levels of IgE and IgG anti-*E.cuniculi* were observed in CC patients with significantly higher rates of positivity compared to the healthy control group. The high seroprevalence and prevalence of microsporidia in tissues of patients with CC would suggest a relationship between microsporidia infection and the etiopathogenesis of CC.

## Figures and Tables

**Figure 1 cancers-14-05342-f001:**
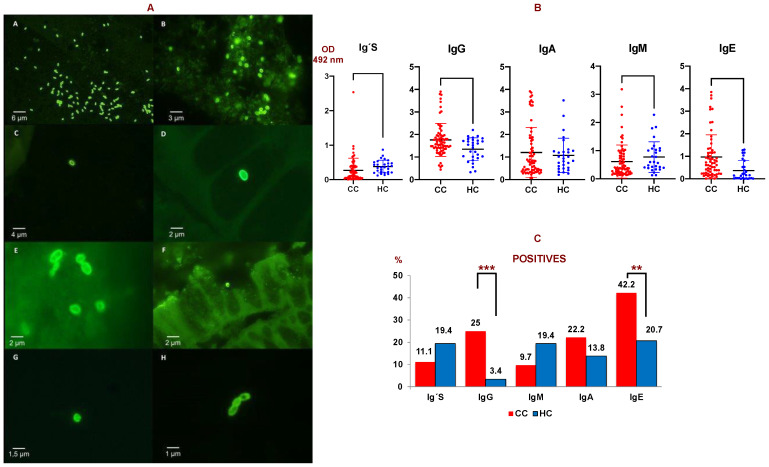
(**A**) IFAT of microsporidia species. A: Positive control of MAb 2C2 with purified spores of *E. cuniculi* (40×); B: Positive control of MAb 6E52D9 with spores of *E. bieneusi* (100×); C: Spores of *Encephalitozoon* sp. in positive biopsies by IFAT with MAb 2C2 (40×); D and E: Spores of *Encephalitozoon* sp. in positive biopsies by IFAT with MAb 2C2 (100×); F: Spore of *E. bieneusi* in positive biopsies by IFAT with MAb 6E52D9 (40×). G and H: Spores of *E. bieneusi* in positive biopsies by IFAT with MAb 6E52D9 (100×). (**B**) Anti-*E. cuniculi* antibody levels (O.D. = optical density) measured by ELISA in colon cancer patients (CC, *N* = 87) and healthy controls (H, *N* = 25). Ig’s: total immunoglobulins. (**C**) Percentage of serum samples considered as positive against *E. cuniculi* in colon cancer patients (CC) compared to healthy controls (HC) obtained by ELISA. Ig´s: total immunoglobulins. Double T bars denote standard deviation (*** *p* < 0.001, ** *p* < 0.01). A Mann-Whitney U test was used.

**Table 1 cancers-14-05342-t001:** Characteristics of patients with Colon Cancer (CC) (*N* = 87).

	*N* (%)		*N* (%)
Gender		Histologic sample type	
Male	56 (64.4)	Adenocarcinoma (AC)	67 (77.0)
Female	31 (35.6)	Mucinous AC	14 (16.1)
Tumor Stage		Infiltrant on adenoma AC	6 (6.9)
I	29 (33.3)	Grade	
II	26 (29.9)	Low grade	47 (54.0)
III	29 (33.3)	High grade	40 (46.0)
IV	3 (3.4)	Invasion	
History of cancer	11 (12.6)	Subserosa	39 (44.8)
Family history CC	13 (29.5)	Muscular	28 (32.2)
Recurrent	6 (6.9)	Submucosa	13 (14.9)
Polyposis	50 (61.0)	Serosa visceral	4 (4.6)
Synchronous cancer	6 (6.9)	Neighboring organ	3 (3.4)
	Mean ± SD *		
Age	69.5 ± 11.5	Vascular invasion	35 (40.2)
Analyzed lymph nodes	17.3 ± 7.4	Lymphatic invasion	35 (40.2)
Lymph node metastasis	1.3 ± 2.1	Perineural invasion	5 (5.7)

* SD: Standard Deviation.

**Table 2 cancers-14-05342-t002:** Microsporidia species detection in CC patient tissue (*N* = 87).

	Tumoral Tissue *N* (%)			Healthy (Peritumoral) Tissue *N* (%)			Tumoral and Healthy (Peritumoral) Tissue *N* (%)		
	PCR	IFAT	Both Techniques (PCR & IFAT)	PCR	IFAT	Both Techniques (PCR & IFAT)	PCR	IFAT	Both Techniques (PCR & IFAT)
*Encephalitozoon hellem/intestinalis*	3 (3.5)	-	-	3 (3.5)	-	-	0 (0.0)	-	-
*Encephalitozoon cuniculi*	3 (3.5)	-	-	3 (3.5)	-	-	1 (1.1)	-	-
*Encephalitozoon* sp.	6 (6.9)	4 (4.6)	2 (2.3)	6 (6.9)	3 (3.5)	2 (2.3)	1 (1.1)	0 (0.0)	5 (5.7)
*Enterocytozoon bieneusi*	0 (0.0)	2 (2.3)	0 (0.0)	0 (0.0)	0 (0.0)	0 (0.0)	0 (0.0)	0 (0.0)	1 (1.1)
Coinfection	0 (0.0)	1 (1.1) *	0 (0.0)	0 (0.0)	0 (0.0)	0 (0.0)	1 (1.1) **	0 (0.0)	1 (1.1) **
Undetermined	0 (0.0)	0 (0.0)	0 (0.0)	0 (0.0)	0 (0.0)	0 (0.0)	1 (1.1) ***	0 (0.0)	0 (0.0)
Total infected patients by techniques (PCR, IFAT, or both)	6 (6.9)	7 (8.0)	2 (2.3)	6 (6.9)	3 (3.5)	2 (2.3)	3 (3.5)	0 (0.0)	7 (8.0)
Total infected patients by tissue (Tumoral, Healthy (peritumoral), or both)	15 (17.2)			11 (12.6)			10 (11.5)		
TOTAL CC PATIENTS INFECTED	36 (41.3 %)								

PCR: polymerase chain reaction. IFAT: immunofluorescence antibody test. (*) One sample had *Encephalitozoon*/*E. bieneusi* coinfection by IFAT in tumoral tissue. (**) Two samples had *E. cuniculi* and *E. hellem*/*intestinalis* co-infection by PCR and by PCR and IFAT in both tumoral and healthy tissue. (***) One sample could not be identified by PCR.

## Data Availability

The data presented in this study are available in this article.

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
