# Peer review of "Latent Microsporidia Infection Prevalence as a Risk Factor in Colon Cancer Patients"

_cancers, 2022, doi:10.3390/cancers14215342_

Round 1

Reviewer 1 Report

General comment

In this research, the authors investigate the presence of microsporidia in tissues of patients with CC without previous specific anti-tumor treatment. The study is very interesting and can be accepted for publication after addressing minor comments for some points below.

Abstract

The abstract needs to have some data about findings of  correlation, ANOVAanalysis.

Introduction

The introduction needs to be more details and add more references 

 Methods

Line 196: Statistical analysis could be better replaced by Data analysis.

In statistical data section you  should  specify which significance level followed?

 Results

In all figure 1 in the manuscript, please remove the word (figure1) from the  above  of the figure.

In Table 2: the data presented is about using Microsporidia species detection in CC patient tissue). Statistically we may need to know whether there is a significant difference among all these tissues. Please indicated why you did not perform this step?

Conclusions

-The conclusion is so general, please include simple specific results.

Author Contributions:

Please take care of using same font type for all text in this section, check line:406.

References

Please check the number of references you used as you duplicated the reference int end of manuscript.

I also think as long as you did not cite references in the text, no need to mention them in the end of the manuscript even if as “Further Readings’’

Author Response

Reviewer 1

Thanks for your comments

General comment

In this research, the authors investigate the presence of microsporidia in tissues of patients with CC without previous specific anti-tumor treatment. The study is very interesting and can be accepted for publication after addressing minor comments for some points below.

Abstract

The abstract needs to have some data about findings of correlation, ANOVA analysis.

Reply

No correlation analysis (Spearman and Pearson) or ANOVA has been performed. It has not been necessary

Introduction

The introduction needs to be more details and add more references 

 Reply

Have been added more details and more references 

Schistosoma japonicum infection is particularly interesting in our study because it is associated with the development of colorectal cancer. The tumorigenic mechanisms that have been studied involve chronic inflammation, mutations of p53 by the action of toxins, and inhibition of the activity of the immune system, promoting the accumulation of myeloid derived suppressor cells (MDSCs) in lymphoid tissues and in the tumor microenvironment”.  Hamid O, et al. 2019

Microsporidia are obligated intracellular parasite that cause severe infections in immunosuppressed subjects. Nevertheless, it has also been detected in healthy immunocompetent subjects. Ruf B, Sandfort J. et al. 1994

Methods

Line 196: Statistical analysis could be better replaced by Data analysis.

In statistical data section you should specify which significance level followed?

 Reply

Statistical analysis has been replaced by data analysis.

It is specified In line 200-202: “The level of significance was taken as a P value of less than 0.05 (bilateral contrast)”

Results

In all figure 1 in the manuscript, please remove the word (figure1) from the above of the figure.

Reply

The word “Figure 1” has been removed from the figure

In Table 2: the data presented is about using Microsporidia species detection in CC patient tissue).

Statistically we may need to know whether there is a significant difference among all these tissues. Please indicated why you did not perform this step?

Reply

The table is only descriptive, since with a very small "n", the differences would not be significant. In addition, we understand that it is not important to make differences, since a simple description is enough.

Conclusions

-The conclusion is so general, please include simple specific results.

Reply

The conclusion has been modified

“More than forty percent of colon cancer patients have microsporidia infection in their tissues vs. zero percent in the tissues of healthy subjects. A significant increase in IgE and IgG anti-E. cuniculi antibodies in CC patients was observed with a significant increase in IgE and IgG anti-E. cuniculi positives in serum of CC patient’s vs healthy controls. The high seroprevalence and prevalence of microsporidia in tissues of patients with CC would suggest a relationship of microsporidia with the etiopathogenesis of CC”.

Reviewer 2 Report

This study has revealed the relationship of microsporidia with the etiopathogenesis of CC and found a significant increase in specific IgE in patients with CC. Those findings are pretty novel but before publish, some minor points need to be addressed.

1) Scale bar was missing in Figure 1A.

2) Somehow, the numbers are overlapping in Table 1.

3) It easily mislead the readers as there are not clear finding presented in the last paragraph from introduction part.

4) IHC staining of the key markers needs to be done in the CC sample to further support the conclusion.

Author Response

Reviewer 2

Thanks for your comments

This study has revealed the relationship of microsporidia with the etiopathogenesis of CC and found a significant increase in specific IgE in patients with CC. Those findings are pretty novel but before publish, some minor points need to be addressed.

1) Scale bar was missing in Figure 1A.

Reply

Scale bar has been added in Figure 1A.

2) Somehow, the numbers are overlapping in Table 1.

Reply: Table 1 has been modified and corrected

3) It easily misleads the readers as there are not clear finding presented in the last paragraph from introduction part.

Reply: The phrase has been corrected

“The main goal of this study was to investigate the presence of microsporidia in tissues of patients with CC without previous specific anti-tumor treatment”

4) IHC staining of the key markers needs to be done in the CC sample to further support the conclusion.

Reply

The samples used in this study were diagnosed positive for colon cancer using the Hematoxylin-Eosin staining technique, which confirms an alteration in tissue architecture compatible with this type of neoplasm.
